# Imaging-based intelligent spectrometer on a plasmonic rainbow chip

Dylan Tua[1,3], Ruiying Liu[1,3], Wenhong Yang[2,3], Lyu Zhou[1], Haomin Song[2], Leslie Ying ®[1] & Qiaoqiang Gan ®[1,2] ✉

Compact, lightweight, and on-chip spectrometers are required to develop portable and handheld sensing and analysis applications. However, the performance of these miniaturized systems is usually much lower than their benchtop laboratory counterparts due to oversimplified optical architectures. Here, we develop a compact plasmonic "rainbow" chip for rapid, accurate dual-functional spectroscopic sensing that can surpass conventional portable spectrometers under selected conditions. The nanostructure consists of one-dimensional or two-dimensional graded metallic gratings. By using a single image obtained by an ordinary camera, this compact system can accurately and precisely determine the spectroscopic and polarimetric information of the illumination spectrum. Assisted by suitably trained deep learning algorithms, we demonstrate the characterization of optical rotatory dispersion of glucose solutions at two-peak and three-peak narrowband illumination across the visible spectrum using just a single image. This system holds the potential for integration with smartphones and lab-on-a-chip systems to develop applications for in situ analysis.

Optical spectroscopy is one of the most widely used techniques for fundamental research as well as industrial processes. However, benchtop systems are usually bulky, expensive, and mainly designed for laboratory and industrial spectroscopic analysis. In recent years, researchers and major industrial players have shifted focus toward developing miniaturized, portable, and inexpensive spectrometer systems, which can enable many emerging applications for on-site, real-time, and in situ spectroscopic analysis in our daily lives[1]. For instance, 195 colloidal quantum dot filters with different optical transmission properties were placed on top of a smartphone camera chip[2]. By processing the large set of sensor readings, this chip-scale system can reconstruct the spectral features of incident light in the visible to near-infrared (IR) spectral range. Another pioneering work employed a single compositionally engineered nanowire as the key active element of an ultra-compact spectrometer chip[3]. Combined with extended post-data processing algorithms, the spectral response of the compact chip can be used to reconstruct the incident spectral information. Over the past decade, various photonic crystal slab filters, plasmonic and metasurface filters were also proposed to be integrated with CMOS camera chips (e.g., refs. 4–6). It was believed that these thin film optical filters can be integrated with each pixel of the camera chip and enable various spectroscopy analysis functionalities, including miniaturized spectrometers (e.g., ref. 7), polarimetric sensing/imaging (e.g., ref. 8) and compressing spectroscopic sensing[9]. However, due to the oversimplified optical design and mechanical limit of compact architectures, the actual spectral identification performance of miniaturized spectrometer systems is usually much lower than their benchtop counterparts.

A strategy to address these limitations is to implement deep learning (DL) in the data processing steps in photonic methodology[10–13]. DL offers much potential to the miniaturization of modern technologies for several reasons. First, it has the ability to exploit information from data that may be indiscernible by traditional methods. Second, its flexibility in design makes it compatible with

[1]Electrical Engineering, University at Buffalo, The State University of New York, Buffalo, NY 14260, USA. [2]Material Science Engineering, Physical Science Engineering Division, King Abdullah University of Science and Technology, Thuwal 23955-6900, Saudi Arabia. [3]These authors contributed equally: Dylan Tua, Ruiying Liu, Wenhong Yang. ✉e-mail: qiaoqiang.gan@kaust.edu.sa

nanophotonic platforms, such as metasurfaces and plasmonic nanostructures[14]. Third, DL algorithms can be applicable to various functions, such as spectral reconstruction[15,16], high-resolution imaging[15,17], classification[18,19], noise suppression[20], and inverse design of photonic structures[15,21–24]. However, DL algorithms in these pioneering efforts are often limited to a single function (e.g., refs. [16,18–20]). This is attributed to the data that are available to train and test these models, which are actually limited by the information contained by the data collected from optical systems. For example, in ref. [16], a spectral encoding chip composed of 252 plasmonic nanohole arrays was used to train a DL reconstruction algorithm. Due to the simplistic design of the plasmonic arrays, the encoding chip was only able to extract information about the spectral peaks of incident light. Thus, other features like polarization were rendered as lost information, limiting the feasibility of the system in spectroscopic applications. This presents an underlying challenge in expanding the capabilities of compact systems enabled by DL. Under these pretenses, physical data with multi-dimensional features is the key to enable the development of more powerful DL-based systems. Consequently, engineering the physical layer (i.e., optical systems[25] and plasmonic/metamaterial nanostructures[26]) to provide more distinguishable training and testing data for DL algorithms has become an exciting and emerging topic to create applications for future artificial intelligence (AI) sensing systems that were impossible for conventional systems[27].

Here we report an intelligent on-chip spectrometer by integrating an on-chip rainbow trapping phenomenon with a compact optical imaging system. Our results show that the plasmonic chip can distinguish between different illumination peaks across the visible spectrum (470–740 nm). Making full use of its wavelength-sensitive structure, the chip can illustrate varying plasmon resonance patterns based on the peaks of the illumination spectrum. By expanding the chip to its 2D structure, the increased complexity of the resonance patterns offer an added level of information in terms of the incident light polarization. By training the DL algorithms with images of the spatial and intensity distributions of the on-chip resonance patterns, spectroscopic and polarimetric analysis is achieved within the same system, respectively. Using a chiral substance (i.e., glucose), which introduces optical rotation to transmitting light, we demonstrate the feasibility of the proposed spectrometer in the sensing of optical rotatory dispersion (ORD), a polarization-specific feature that is useful for detection and quantification of chiral substances. Analysis performed by the DL model shows that the algorithm is capable of accurately predicting the optical rotation introduced by glucose based on the resonance pattern of the plasmonic chip. This performance is preserved even when analyzing resonance patterns under illumination of multiple peaks. This image-based spectrometer enabled by DL is capable of performing both spectroscopic and polarimetric analysis by utilizing a single image of the nanophotonic platform. As such, our proposed system is empowered with a far-reaching impact on spectropolarimetric sensing applications.

## Results

### 1D rainbow chip

Here we will first employ the rainbow trapping effect[28–31] to develop an on-chip spectrometer system (e.g., refs. [32–34]). As proof of concept, Fig. 1 illustrates the proposed system and the design of the 1D rainbow chip. Wavelength splitting functionality can be realized by the plasmonic chirped grating (Fig. 1a). The geometry of this surface grating changes gradually, resulting in the spatial tuning of the local plasmonic resonances (i.e., so-called trapped "rainbow" storage of light[28,33]). As shown in Fig. 1b, we employed focus-ion milling to fabricate a chirped grating on a 300 nm thick Ag film. We intentionally assembled graded 6-groove units with varying period changing from 244 to 764 nm. The width of the grooves are 200 nm (see Note S1 for more details on the fabrication). Under the normal incidence of a white light, one can employ a simple reflection microscope system (Fig. 1c) to observe obvious "rainbow" color images (top panel in Fig. 1d) due to the plasmonic resonances supported by these gratings (Note S2 in the Supplementary Information). By introducing narrowband incident light to illuminate the sample (see Fig. 1e for the spectral lineshapes of varying wavelengths), one can distinguish these different wavelengths from their spatial patterns (lower panels in Fig. 1d). Based on these spatial pattern images, a one-to-one correspondence of the resonant pattern can be established with the incident wavelength, indicating the

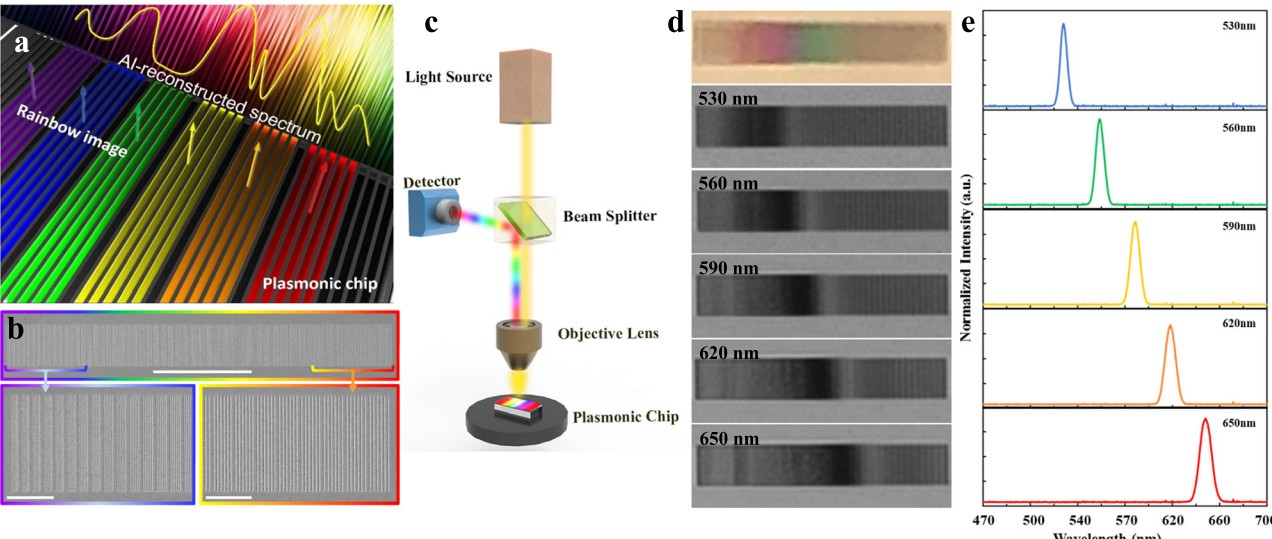

**Fig. 1 | Plasmonic chirped gratings for on-chip spectrometer. a** Illustration of the plasmonic chirped grating being used as an on-chip spectrometer. **b** SEM images of the plasmonic grating fabricated on a thin Ag film via focus-ion beam milling. The grating is composed from an assembly of multiple 6-groove units. The period of the units increases from 244 to 764 nm along the length of the chip. This allows for a spatial distribution of plasmon resonances, i.e., a "rainbow trapping" effect. The scale bar for the top image is 50 μm; the scale bars for the bottom two images are both 10 μm. **c** Schematic of a simple reflection mode microscope system to observe "rainbow trapping" patterns on the plasmonic chip. **d** Images of the "rainbow" color pattern produced by the chirped grating (top panel) and spatial patterns of the plasmonic resonances for narrowband illumination at several center wavelengths (lower panels). **e** Spectral lineshapes for the spatial patterns of the corresponding center wavelengths in (d).

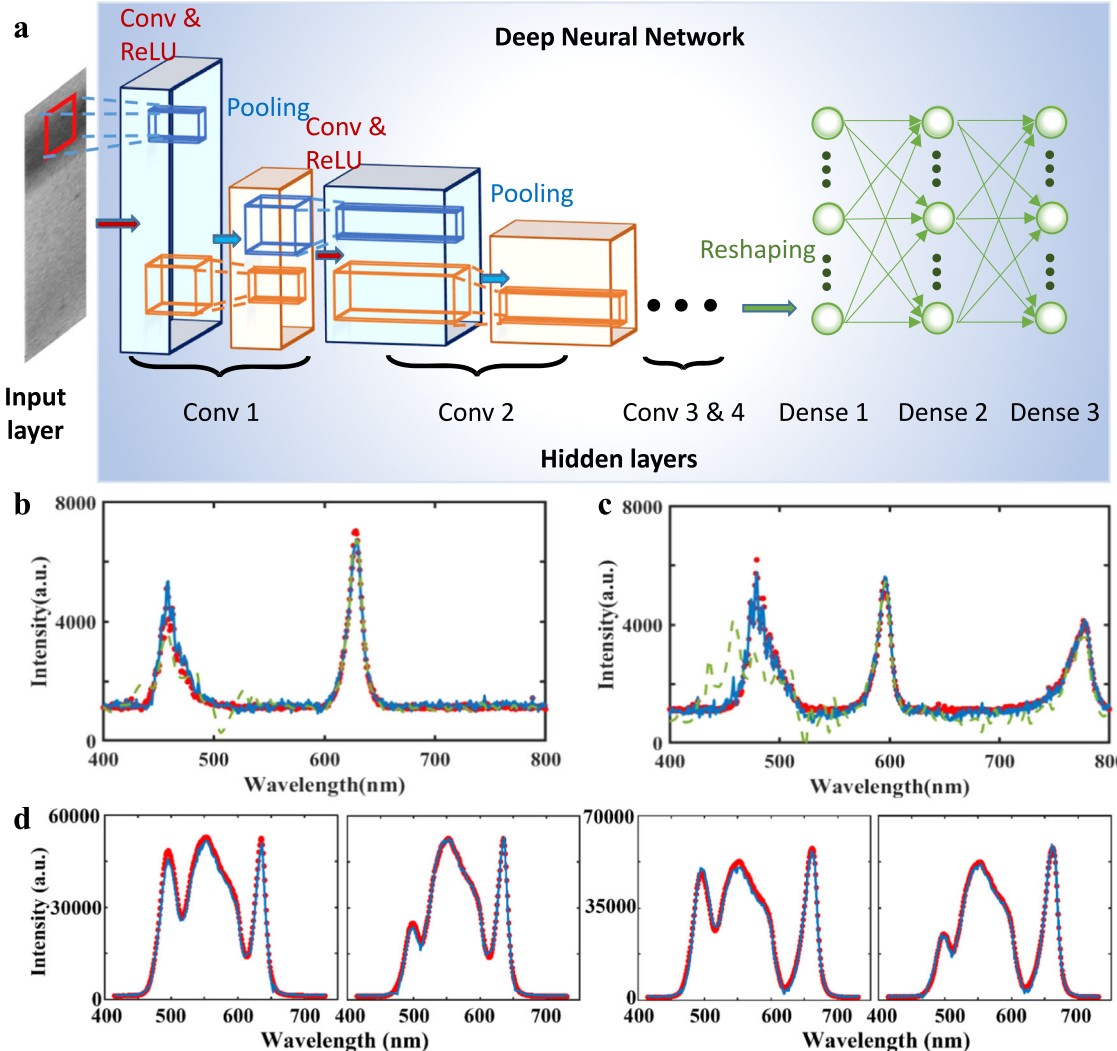

**Fig. 2 | DL-based reconstruction. a** The architecture of the deep-learning network. For the case of **b** two-peak and **c** three-peak wavelength combinations, comparison of deep-learning-reconstructed spectrum (solid blue line), spectrum calculated using conventional method (dashed green line), and the gold standard spectrum measured by a conventional spectrometer (dotted red line). **d** Reconstruction of broadband spectra introduced by three LEDs with their respecitve spectra overlapped with each other. Red dots are measured spectra. Solid blue lines are reconstructed spectra.

foundation of an on-chip spectrometer (e.g., ref. 32). As such, we investigate the capabilities of the proposed system in observing a spatial correspondence for arbitrary spectral features. Using DL-assisted data processing and reconstruction methods, this wavelength splitting functionality can enable an intelligent and miniaturized spectrometer platform for optical integration.

## DL-based reconstruction

Accurate spectrum reconstruction is one of the most important procedures required by miniaturized spectrometer systems, which, however, contains major challenges in previously reported works. For instance, in the recently reported single nanowire spectrometer[3], the spectral pattern was measured for each of the $n$ photodetector units. A linear equation is formed and solved based on the spectral pattern and the pre-determined spectral response function, whose solution gives the reconstructed spectrum (see Note S3 for details of how to reconstruct the spectrum)[3,35,36]. However, as with all linear methods, the reconstructed target spectrum can be largely distorted when there is measurement noise and/or errors in the pattern image. Despite a number of methods to address the issue of ill-posedness, such as adaptive Tikhonov regularization and iterative algorithms such as

compressed sensing[37], these methods heavily rely on the accuracy of the estimated spectral response function, which is typically not guaranteed. In addition, the regularization, which involves tedious parameter tuning, can introduce bias to the reconstructed spectrum. The computational complexity can be high when solving a large number of equations. Due to the above limitations of the existing spectra reconstruction methods, there are visible deviations from the actual spectrum for the existing miniaturized spectrometers (e.g., see Note S3 for examples of this deviation). Here we employ a DL-based method to address all of the above-mentioned challenges. Specifically, we propose the concept of an intelligent rainbow plasmonic spectrometer driven by DL and build an example of such a spectrometer with plasmonic chirped gratings (Fig. 2). The spectrometer predicts the unknown incident light spectrum from the measured resonance pattern image using a deep neural network, bypassing the traditional linear model using response functions.

The architecture of the experimental system is shown in Fig. 2a. The intelligent spectrometer contains three main parts: spatial pattern, pretrained neural network, and corresponding wavelength. The spatial pattern which is the reflection image of miniaturized rainbow spectrometer (Fig. 1a) captured by reflection microscope system (Fig. 1c), is

a unique fingerprint of the incident light, and thus is used as the input to the neural network (Fig. 2a). By using DL, we aim to exploit and generalize the intricate relationship from the spatial pattern to the incident wave for a specific plasmonic chirped grating, such that the pretrained neural network is able to predict the intensity, wavelength, and polarization of incident light accurately. As shown in Fig. 2a, our deep neural network consists of a set of input neurons that are interconnected to a number of neurons in hidden layers. Information propagates forward via a linear operation such as convolutions with an activation function often seen as a rectified linear unit (ReLU), followed by a nonlinear pooling operation in a pooling layer. Several convolutional layers and pooling layers are stacked and the final output is obtained by flattening the output of the last pooling layer via dense (fully connected) layers. In other words, each reflection image as neural network input would get a light spectrum as a prediction output. Before putting the neural network into use, the synaptic strengths between each layer (i.e., the weights of the linear operation) need to be adequately trained via a back-propagation algorithm such as gradient descent or adaptive optimizer.

During training, a fiber-coupled LED light (cool LED, PE-4000) was employed as the incident light with the option to combine different wavelengths, as shown in Fig. 2b. Here we first combined pairs of two and three arbitrary wavelengths (e.g., $525 + 660$ nm and $435 + 460 + 595$ nm) with arbitrary intensities as the incident light to illuminate the chirped plasmonic grating. Reflection images of resonance patterns were captured by the ×4 microscope system. A total of 500 spectra with different peaks and intensities and images of their corresponding resonance patterns were obtained. The spectra were used as the targeted outputs (i.e., desired reconstructions) of the training data, while the images were used as the inputs. This was not only used to train the neural network, but also to calibrate the spectral response function for the conventional method used in ref. 3. We obtained another 100 spectra with different peaks and intensities beyond the scope of the training data for testing the proposed method and conventional method in ref. 3. Mean square error was used to represent a loss function between the normalized and desired output, and the loss of the training set was used to generate gradients (pure learning). The hyperparameters (for example, number of hidden layers, neurons, and learning rates) were set according to the performance on the validation set. A convolutional neural network with four convolutional layers and two fully connected layers with a total of 600 neurons was selected (see Note S4). Figure 2b and c shows the results using two sets of testing data (peak wavelengths at $460 + 635$ nm and $470 + 595 + 770$ nm, respectively) not included in the training process. The dotted red lines show the gold standard spectrum of the incident light measured by the conventional grating-based spectrometer. The solid blue curves are the reconstructed spectra, agreeing very well with the actual spectrum. In contrast, we also calculated the spectrum using conventional methods[3] based on the same 500 sets of training data and plotted the spectrum by the dashed green line. One can see that the spectral features near 460 and 470 nm were obviously misinterpreted. The result demonstrates the proposed intelligent imaging-based spectrometer on a chip is credible and applicable in this scenario (see Note S4 for comparison of the AI and conventional processing methods).

Reconstruction of arbitrary spectra will require sufficient training data to cover various spectral features of different spectral samples. In particular, one needs to collect combinations of different narrowband and broadband spectra. As a preliminary proof-of concept, we employed the LED light source to demonstrate a broadband spectrum reconstruction. This LED light source allows for a combination of multiple LEDs to construct more complicated spectra. As a result, the spectral feature is different from individual LEDs, especially at the overlapped regions among different LED spectra. For the training dataset, we collected individual, double-wavelength and triple-

wavelength combinations (see Note S5, Table S1). After that, we collected four different sets of three-wavelength combinations with different intensities for testing, which were not included in the training datasets (see parameters in Table S2). Figure 2d shows four representative reconstructed spectra (blue solid lines). Compared with the measured spectra (red dots), one can see that the spectral features (especially the feature at the overlapped regime) were well predicted. In principle, the procedure for arbitrary spectrum reconstruction will follow the same practice but will need more training to include possible features in the target spectra, which is still under investigation.

On the other hand, spectral resolution is one of the most important parameters to evaluate the performance for conventional spectrometers. Here we employed a broadband halogen lamp through a liquid crystal filter to reveal its resolution in wavelength shift. We first captured 10,000 images of the rainbow chip under the illumination of narrowband incidence from 600 to 650 nm with the step size of 0.1 nm tuned by the liquid crystal filter. Their actual spectra were characterized using the fiber-based spectrometer. 8000 (and 9000) images have been selected randomly as training data. After training, we tested the remaining 2000 (and 1000) images which were not included in the training data. As shown in Fig. 3a and b, single peaks can be reconstructed and well resolved with the peak shift of 0.5 nm (Fig. 3a) and 0.2 nm (Fig. 3b). The accuracy of the reconstructed peak position is 87–95% for the peak shift of 0.5 nm, and 81–90% for the peak shift of 0.2 nm (see Table S3). More technical details to resolve wavelength shifts with different step sizes are listed in Note S6.

To further reveal the spectral analysis capability, we then introduced two narrow peaks controlled by a programmable acoustic optical filter to illuminate the grating simultaneously. Seven representative spectra of the incident narrowband light are plotted in Fig. 3c: One peak was fixed at the wavelength of 596.8 nm. The other narrow peak was tuned from 596.8 to 646.8 nm with the step size of 0.1 nm. As shown by spheres in Fig. 3d, these two adjacent incident peaks produced a combined spectrum, showing that the two peaks gradually separate apart with each other and therefore can be resolved by the conventional spectrometer. In this experiment, we collected 901 images as the training set and 100 images for testing (see Table S4, check Note S7 for more details). The reconstructed spectra are plotted by solid curves in Fig. 3d, agreeing perfectly with the measured spectra. One can see that the two-peak identification is similar to determining the optical resolution in imaging applications using the Rayleigh criterion[38]. According to our reconstructed and measured spectra, the two-peak feature was clearly resolved when the wavelength difference is beyond 2 nm (see detailed analysis in Note S8). These preliminary data indicated the potential using the smart rainbow chip system to perform high-resolution spectral analysis with the equivalent performance compared with conventional spectrometers. Next, we will extend the 1D grating into 2D to enable polarimetric spectroscopy using the compact smart system, which is superior over conventional optical spectrometer systems.

## Polarimetric spectroscopy using a 2D rainbow chip

Polarization is one of the most fundamental properties describing the path traversed by the electric field vector of an optical beam. Polarization-sensitive coloration phenomenon has been observed in many animals' skin (e.g., ref. 39), indicating the potential application in biomimetic optical communication. In addition, polarimetric sensing and imaging techniques are widely used in material characterization, remote sensing and imaging, and security and defense applications. For instance, a compact polarimetric imaging system was reported using a large-scale dielectric metasurface component (i.e., 1.5 mm in diameter used in ref. 40) in the regular imaging system. Multiple polarizer elements and optical coupling elements can therefore be simplified, compactifying the footprint of entire optical systems relying on conventional polarization optics. Miniaturization and

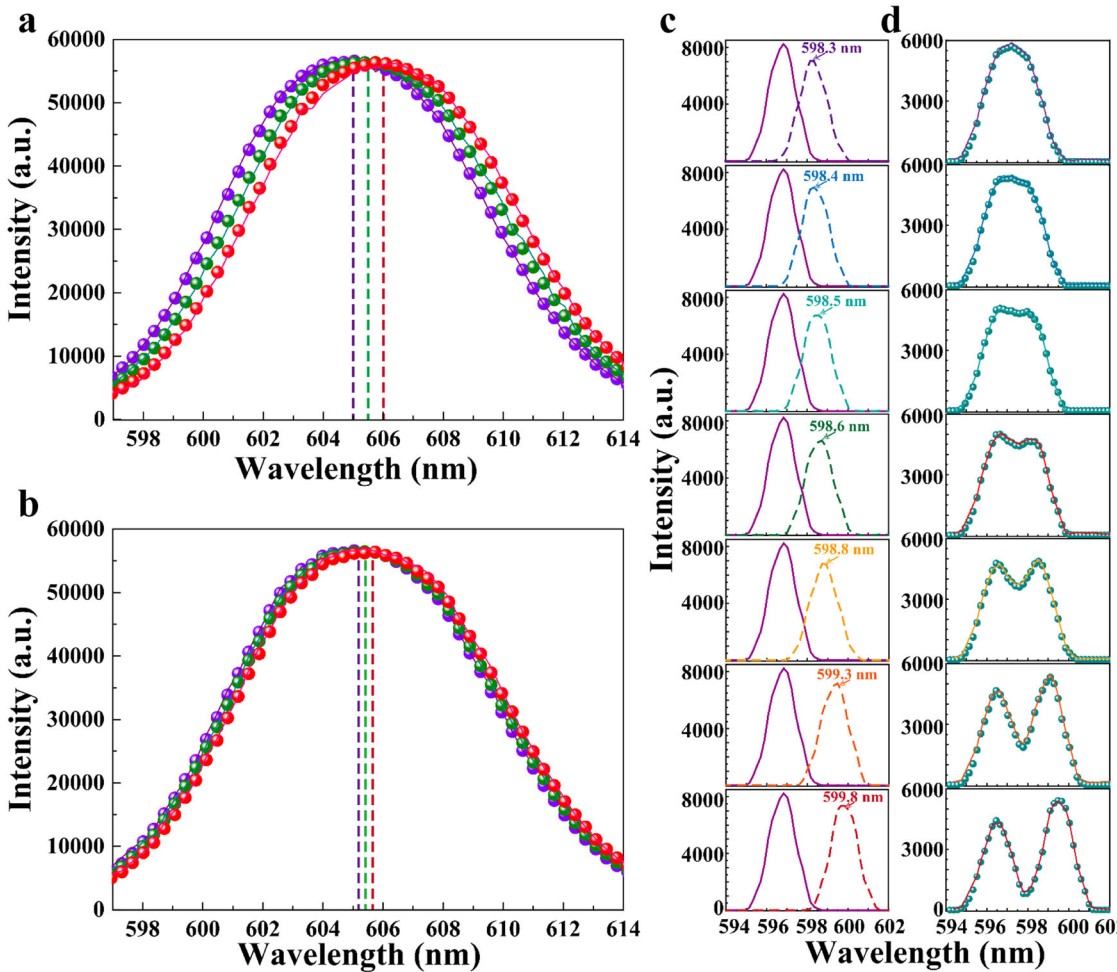

**Fig. 3 | The resolution of the smart system. a, b** DL-reconstructed spectra (solid lines) with a step size of 0.5 nm (**a**) and 0.2 nm (**b**), respectively, and the measured spectra using a conventional spectrometer (spheres). The peak positions are indicated by vertical dashed lines [i.e., 605.0 nm (purple line), 605.5 nm (green line), and 606.0 nm (red line) in (**a**), and 605.2 nm (purple line), 605.4 nm (green line), and 605.6 nm in Fig. 3b]. **c** The spectra of the two incident light measured independently. **d** DL-reconstructed spectrum (solid lines) and the measured spectra of the two combined peaks using a conventional spectrometer (spheres).

simplification of conventional, bulky, and time-consuming optical characterization represent an emerging and important research topic[1]. Here we demonstrate that the plasmonic rainbow chip spectrometer can introduce a simplified, compact, and intelligent spectropolarimetric system with accurate and rapid spectral analysis capabilities. Figure 4a shows a 2D grating with graded geometric parameters (see Note S9 for fabrication details and Note S10 for numerical modeling results in Fig. S9). The period of the grating varies from 439 nm (Fig. 4b) to 739 nm (Fig. 4c) in two directions. By capturing the reflection image of this 2D chirped grating, one can see a "cross" bar with two arms representing two polarization states (see reflection images at four different wavelengths in Fig. 4d and more images in Fig. S10 in Note S11 to determine the spectral range of this chip). Intriguingly, the intersection position (indicated by the white cross marks) corresponds to the peak position of the incident wavelength, and the intensities of the two arms represent the component intensities of the two polarization states along the horizontal and vertical directions, respectively. Following a similar data training process using different combined LED lights, an intelligent polarimetric spectrometer is demonstrated (see Note S12 for details on the reconstruction method, Note S13 for explanation of the training and testing procedure, and Table S5 for the parameters of both datasets). Figure 4e and f shows the reconstructed spectra using two sets of testing data (two different polarized light with peak wavelengths at 490, 595, and

635 nm, respectively) not included in the training process. One can see the reconstructed spectra (dark blue curves) agree very well with the measured spectra (red curves). Intriguingly, this unique intelligent spectrometer chip can enable rapid polarimetric spectroscopy sensing applications. Next, we will demonstrate simple and intelligent characterization of ORD using this 2D polarimetric spectrometer chip.

## ORD characterization using the 2D rainbow spectrometer

Conventional ORD systems measure the optical rotation introduced by a substance as a function of the incident wavelength (as illustrated in Fig. 5a). To perform accurate characterization, special facilities are usually required with multiple polarization generators and analyzers (i.e., so-called polarimetry systems[41]). By scanning the illumination spectrum and comparing its output polarization state to its initial polarization state, one can obtain the ORD of the sample. The accuracy in determining the ORD depends on the polarizer tuning resolution. Manually tuned polarizers require fine rotation to get the complete spatial distribution for a single wavelength, which is tedious and time-consuming in experiment. They are also inaccurate due to errors introduced during measurement (e.g., parallax error). Faster and more accurate measurement is achievable using electronically tuned polarizers. However, these polarizers are costly and require periodic recalibration to maintain optimal performance. In contrast, the proposed imager-based system can provide broadband spectral information and

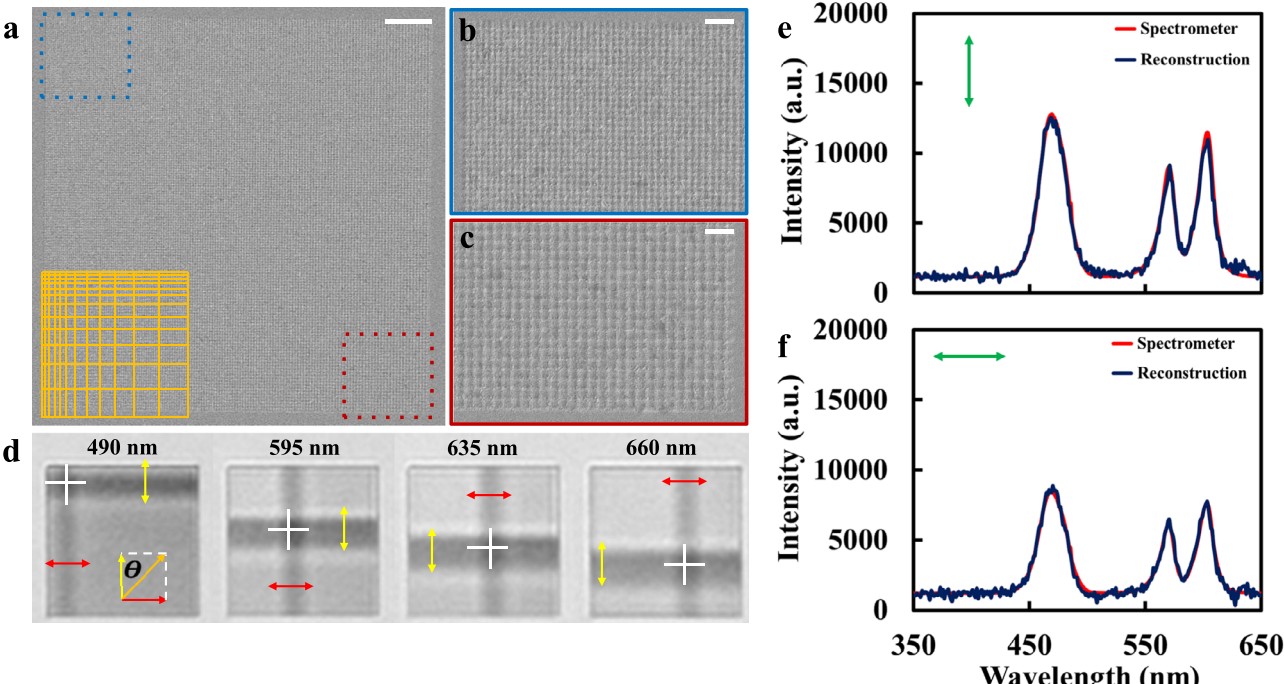

**Fig. 4 | 2D chirped grating for polarimetric spectroscopy. a** SEM image of a preliminary sample with the period varying from 439 to 739 nm along two directions. Scale bar is 10 μm. **b, c** Zoom-in images at two corners of the chirped grating. Zoom-in regions correspond to squared off areas of matching color in (**a**). Scale bars are 2 μm. **d** Reflection images of the grating under the illumination of 490, 595, 635, and 660 nm light. **e, f** Reconstructed spectra of vertically polarized (**e**) and horizontally polarized (**f**) light with peak wavelengths at 490, 595, and 635 nm. Dark blue lines are reconstructions from the proposed intelligent spectrometer, while red lines resemble reconstruction from a conventional benchtop spectrometer.

polarization distribution from a single image. Therefore, the time-consuming spatial rotation and wavelength scanning processes can be reduced significantly in our 2D imaging characterization.

As a proof-of-concept, here we demonstrate how the 2D grating can be used as a spectropolarimetric system for glucose sensing applications. For conventional spectropolarimetric characterization, it is important that the system be able to accurately measure the ORD of a sample across a broad spectral range. In addition to the issues stated above, conventional systems (Fig. 5b) require tunable narrowband illumination sources to measure the optical rotation for one spectral peak at a time. However, this method is time-consuming and adds to the large amount of tuning already required by the polarizers. Moreover, this approach adds further constraints to the system, as its spectral resolution and operating range become dependent on the tunability of the narrowband illumination source. Our imager-based system enables the optical rotation measurement under the illumination of multiple spectral peaks at once by training the DL algorithm with images of the graded grating under illumination with multiple peaks (see Note S14 for more details). Such capability would allow for more thorough and efficient analysis as well as the use of broadband illumination sources.

In our imager-based setup (Fig. 5c), we kept the first polarizer for fixing the polarization state of the incident light. Conversely, we replaced the analyzer with a beam splitter located between the plasmonic grating and camera to observe the reflection mode of the chip. For the light source, we continued to use the fiber-coupled cool LED. A grayscale camera attached to an optical microscope was used to observe the cross-bar patterns on the chip. For the DL reconstruction, our training data consisted of 26,100 images of the graded grating under various illumination conditions (see Table S6 in Note S15). As shown by examples in Fig. 5d, we captured a wide variety of cross-bar resonance patterns. Air and deionized (DI) water were the samples used for capturing these training data. The trained DL model was then

tested using 540 images of the chip under similar illumination conditions (Table S6). Testing images were captured using aqueous glucose solutions of 2, 10, and 30%. Under the same incident polarization, light-matter interactions with glucose will result in a different output polarization of the illumination spectrum than those with air or water. Due to the wavelength-dependent spatial distribution of the cross-bar patterns on the grating, multiple patterns can be created for each peak in the illumination spectra at once. The DL network can then predict the spectral peaks and their respective polarization states, corresponding to each pattern (see Note S15 for more details on the data training and testing procedure).

To demonstrate the multi-spectral sensing capabilities of our imager-based system, we collected an additional set of training and testing data under double-peak and triple-peak illumination (see Table S7 and Table S8 for parameters of these datasets, respectively). Figure 5e illustrates images of the chip under both of these conditions. To make the spectral peaks as individually noticeable as possible, we selected 525 and 660 nm for the double-peak illumination and 470, 595, and 740 nm for the triple-peak illumination (indicated by red arrows in the left panel). Figure 5f and g plots the double-peak and triple-peak predictions, respectively, of the DL model for 2, 10, and 30% aqueous glucose solutions. The deviations of reconstructed polarization angles (50 data at each wavelength) range from 0.07° to 0.45° (see details in Table S9). For direct comparison, we plotted solid (2–30%) and dashed (0%) curves representing experimental measurements of the ORD, derived through conventional methods (Fig. 5b), for each of these solutions (see Note S16). One can see that the predictions made for both illumination conditions agree well with their respective ORD curves, indicating that the DL algorithm was able to predict the optical rotation introduced by each of the glucose solutions. Moreover, not only did the algorithm identify the peaks of the illumination spectrum, it also isolated them to analyze their unique polarization states despite them all being fixed to the same incident

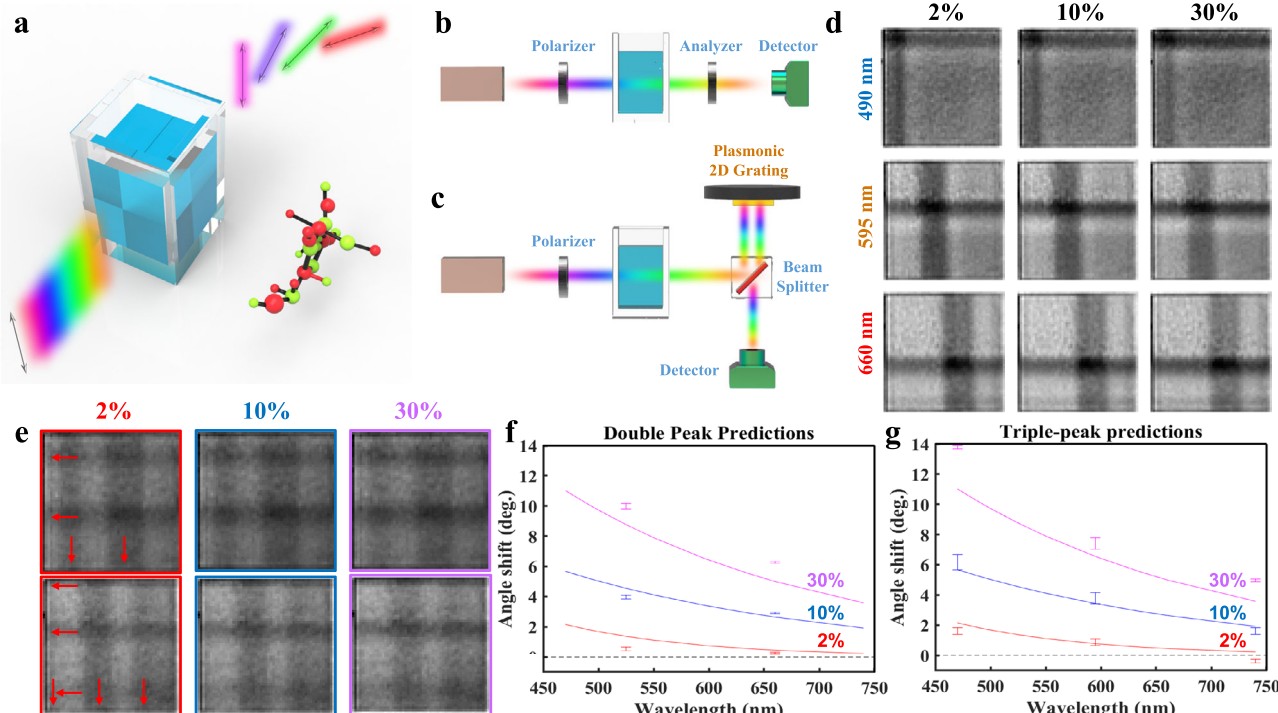

**Fig. 5 | Simpler, accurate, and intelligent spectropolarimetric analysis.**
**a** Illustration of optical rotatory dispersion when light passes through a chiral substance such as glucose. **b, c** Setups for spectropolarimetric analysis using **b** a traditional system and **c** our proposed image-based graded grating system. Our proposed system replaces the second polarizer (analyzer) from the conventional system with a beam splitter and our plasmonic grating. A CCD camera captures images of the plasmonic grating reflection mode, in which the cross-bar patterns appear as a set of dark bars formed from the coupling of light with the grating. **d** Images of the dark bar patterns under single-peak illumination, for 490, 595, and 660 nm illumination with 0% (DI water), 10%, and 30% glucose solutions. Although difficult for the human eye, the deep-learning algorithm can detect differences in the intensities of the horizontal and vertical dark bars based on the concentration of the sample. **e** Images of the dark bar patterns produced under double-peak (top row) and triple-peak (bottom row) illumination. When multiple peaks are used in the illumination spectrum, multiple patterns can be produced at once. **f, g** Predictions of the optical rotation, represented as data points (error bars indicate the standard deviation of measureddatasets), introduced by various glucose solutions for **f** double-peak and **g** triple-peak illumination. The solid curves represent the ORD derived from the experiment using conventional methods. Data were collected for 2% (red), 10% (blue), and 30% (purple) glucose concentrations.

polarization. In contrast, pure water solution (i.e., 0%) did not introduce any rotation (i.e., the black dashed line). As such, our imager-based system can perform rapid spectroscopic and polarimetric analysis of chiral samples, which is essential for on-site real-time and point-of-care applications. It should be noted that in this proof-of-principle study, only 29 different angles with a step size of 1.0° were collected as the training data using a manually tuned polarizer (see Table S6–S8). Due to this limited training dataset, the reconstructed ORD shows inconsistency with the measured curves. This limitation is a technical issue which can be improved using finely tuned electronic-driven polarizers to produce training datasets for future studies.

## Discussion

In conclusion, an intelligent, image-based, on-chip spectrometer is proposed and experimentally demonstrated. Gratings with graded 1D and 2D structures were fabricated and investigated for their resonance patterns caused by the surface plasmon coupling of light. Due to the nonuniform spatial and intensity distributions of the grating patterns, different resonance patterns could be observed depending on the spectral peaks and polarization state of incident light (i.e., the dark bar and dark cross-bar patterns on the 1D and 2D gratings, respectively). By exploiting these features of the graded gratings, information about the illumination spectrum can be extracted from simple observation of the on-chip resonance pattern. Intriguingly, a DL algorithm was integrated into the proposed spectrometer system. By training the algorithm with images of various resonance patterns and the lineshapes of their corresponding illumination spectra, spectroscopic analysis was realized. Meanwhile,

polarimetric analysis was achieved by training the algorithm with images of resonance patterns under a broad range of polarization states. Our results show that spectral reconstructions performed by the proposed spectrometer agree well with the spectra measured by a conventional benchtop spectrometer, demonstrating the capability of the proposed system to perform accurate spectroscopic analysis (see Table S10 in Note S17 for comparisons between the rainbow spectrometer and previous works). Spectroscopic analysis was also performed for horizontally and vertically polarized illumination, demonstrating the capabilities of the proposed system in reconstructing the illumination spectra and distinguishing them between both polarization states. Analysis performed by the DL algorithm show that the proposed system is further capable of accurate and timely polarimetric analysis based on the intensities of the cross bars of the 2D grating resonance patterns. Most notably, both spectroscopic and polarimetric analyses are made possible by the proposed system using a single image of the plasmonic platform. Moreover, DL predictions of the ORD introduced by various glucose solutions indicate the capabilities of the proposed system to perform accurate detection and quantification of chiral substances. The image-based design of the proposed spectrometer system removes the need for optical elements, as well as wavelength scanning and rotating processes. Our proposed image-based spectrometer marks the realization of high-performance spectropolarimetric analysis in a single compact and lightweight design, giving it much potential for use of deep optics and photonics[10] in healthcare monitoring, food safety sensing, environmental pollution detection, drug abuse sensing and forensic analysis.

## Methods

### The optical microscope system

Our setup for collecting the training and testing data for all experiments is composed as a simple reflection mode microscope system (refer to Fig. 1c for the general schematic). We use an Olympus IX81 optical microscope as the centerpiece for our setup. The optical microscope features multiple ports and attachments, including an input reflection mode optical chamber to attach a light source to, and two output chambers that are connected to an attachable camera and optical fiber, respectively. The software and hardware setup allow the optical output of the microscope to be controlled between three paths: the first to the eyepiece for direct visualization, the second to the attached output camera chamber, and the third to the output optical fiber connected to a benchtop spectrometer. In our setup, we used a Hamamatsu ORCA-03G grayscale camera and an Ocean Optics Jaz visual spectrometer at the mentioned output ports. For the incident light, we use a pE-4000 cool LED for selective narrowband illumination, and a halogen lamp (Olympus U-LH100L) coupled to a liquid crystal tunable filter for broadband illumination. Chips with the 1D and 2D grating structures were placed on the microscope stage, with their orientation facing downward for analyzing their reflection mode. A ×4 objective lens (NA = 0.13) was used to analyze the structures and their resonance patterns.

### ORD measurement settings

A sample cuvette with an optical path depth of 5 cm was placed between the incident light source and input chamber of the microscope. The cuvette allowed for the placement of water and various glucose solutions in the optical path of the incident light. Finally, a linear polarizer (Thorlabs PRM1) was placed after the cuvette. The polarizer has a rotational scale engraved along the lens with a micrometer attachment for fine-tune adjustment and precise measurement. The polarizer served the dual purpose of controlling the incident light polarization for spectrum reconstruction, while allowing us to analyze the optical rotatory dispersion (ORD) introduced by various aqueous solutions for polarimetric sensing.

### Image collection settings

Slidebook 5.0 software was used to control the optical microscope and grayscale camera. Images of the grating structures were captured while they were placed on the microscope stage. All images were captured at full size (1024 × 1344 pixels) and then cropped down to the smallest size that can capture the entire pattern before being used for training and testing. This resulted in image sizes of 54 × 54 pixels for the 2D grating images (e.g., Figs. 4d, 5d, 5e), and 187 × 34 pixels for the 1D grating images (e.g., Fig. 1d), corresponding to the actual sample size of ~87 × 87 μm and 302 × 55 μm, respectively.

## Data availability

The data that support the findings of this study are available from QG upon request.

## Code availability

The code for the deep-learning model designed for this study is available from L.Y. upon request.

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

## Acknowledgements

W.Y. and Q.G. are sponsored by the baseline of Physical Science Engineering Division, King Abdullah University of Science and Technology (BAS/1/1415-01-01) and the NTGC-AI program (REI/1/5232-01-01).

## Author contributions

Q.G. came up with the concept and proposed strategies for this study. R.L. and L.Y. designed the DL algorithm used for the analyses. D.T., L.Z., and W.Y. designed the patterns for the 1D and 2D graded grating structures, as well as fabricated the rainbow plasmonic chips using both structures. W.Y. and H.S. performed the numerical modeling of the structure. D.T. and W.Y. collected the training and testing data used for modeling the DL algorithm; data training and testing was performed by R.L. D.T. made the aqueous glucose solutions used in this study, as well as performed the experimental measurements of their respective ORD using conventional methods. All authors wrote and reviewed the manuscript.

## Competing interests

The authors declare no competing interests.
