## [Peer Review File · Nature Communications]

Imaging-based intelligent spectrometer on a plasmonic rainbow chipREVIEWER COMMENTS

Reviewer #1 (Remarks to the Author):

In this manuscript, the authors report a miniaturized spectrometer that can detect the spectral and polarization information with the help of trained deep neural networks. First, the authors demonstrate a 1D chirped grating, which shows distinct patterns under different illuminations. By capturing the relationship between the patterns of the microscope images and the spectra, a neural network can predict the spectra from the image patterns. Next, a 2D grating is used to resolve spectra for orthogonal polarization, and based on that optical rotary dispersion can be measured. Overall, the idea of the work is novel, the results are comprehensive, and the manuscript is written very well. I recommend publication of the manuscript in Nature Communications, after the authors address some questions and comments.

1. I suggest the authors citing some relevant papers, including "Probabilistic Representation and Inverse Design of Metamaterials Based on a Deep Generative Model with Semi-Supervised Learning Strategy," Adv. Mater. (2019); "Compounding meta-atoms into metamolecules with hybrid artificial intelligence techniques," Adv. Mater. (2020); "Deep learning for the design of photonic structures", Nature Photonics 15, 77 (2021); "Pushing the limits of functionality-multiplexing capability in metasurface design based on statistical machine learning," Adv. Mater. (2022); "Ultraspectral Imaging Based on Metasurfaces with Freeform Shaped Meta-Atoms," Laser & Photonics Reviews 16, 2100663 (2022).

2. I suggest the authors making more detailed comparisons between the current work and the literature (including but not limited to Ref. 4-6 and "Ultraspectral Imaging Based on Metasurfaces with Freeform Shaped Meta-Atoms." Laser & Photonics Reviews 16, 2100663 (2022)), regarding the approach/design, device size, spectral resolution, realized functionalities, etc. A table would be helpful.

3. On page 5 line 9, the authors mentioned 'A fully connected network with 4 convolutional layers'. People typically distinguish the terms convolutional neural networks (CNNs) and fully connected neural networks (FCNN) since they have different configurations and data flows.

4. In the first part of the work, the authors use a 1D chirped grating. If the length of the groove or grating is very long and fabrication is nearly perfect, the pattern should only change in the horizontal direction (where the period is changing), maintaining nearly the same value in the vertical direction. Is this the case in this work? If so, it might be advantageous to average the image along the vertical axis in order to convert the image from a two-dimensional (2D input) format to a one-dimensional (1D) format. The neural network can then be built using an FCNN (or 1D CNN), which will be more efficient in terms of training and prediction.

5. The so-called one-to-many mapping issue often exists in the inverse design problem. For instance, very similar or almost identical spectra maybe produced by distinct structures. This issue causes challenges in the inverse design. Does the current study encounter the one-to-many mapping issue? In other words, do apparently dissimilar spectra result in a very similar or almost identical pattern (limited by the resolution of the camera)? If so, what approach did the authors take to resolve or mitigate the issue?

6. The present work is in reflection configuration rather than transmission configuration. Is there a special reason, and what are the possible challenges in transmission-type devices using chirped gratings?

Reviewer #2 (Remarks to the Author):

Efforts to miniturize spectroscopic systems are currently a very timely area - but such miniturization usually leads to inaccurately reconstructed spectra. Attaining accurate reconstruction by simply exploiting images of spectral resonances is a key current and ongoing objective of the broader field of optical spectroscopy.

What this paper reports is a novel artificial-intelligence based method for obtaining accurate such spectra - for miniaturized devices. Although the training time for the AI method may be long, the time for spectrum-reconstruction is efficient once the overall model is calibrated.

The 'rainbow trapping' structure that the authors use for obtaining the images has been optimized and pioneered by the authors over the years (e.g., Refs. 24-26]); it allows for localizing different incident light-wavelengths to different spatial points along the structure (a chirped plasmonic grating), thereby being very broadband and obviating the need for (many slightly different) resonant filters.

Overall, this is a well-written, high-quality work, combining nicely several timely areas (slow light / 'rainbow trapping', optical spectroscopy and polarimetry, and AI-based systems) to attain a useful functionality (described above).

As such, it makes an interesting / novel contribution to the broader field, meriting publication in NCOMMs after, from my perspective, the authors also address the following points:

1. Is the used AI method guaranteed to converge to a global (rather than 'only' local) error-minimum point? That would be interesting to know, in order to assess the generality and general-reliability of the method (here, it does appear to be ok).
2. What is the maximum spectral resolution that can be attained with this scheme (on the reported structure)? Also, what is here the (well-known) figure-of-merit for the reported spectro-polarimeter?
3. How much susceptible (or not) to noise is the scheme, and why?
4. It should be interesting for the journal's broad readership to compare this scheme with other standard schemes / papers (e.g., cf. Refs. [1-4]) for spectral and polarimetric analyses (e.g., in terms of fabrication requirements, scalability, etc) in order to even more clearly outline and emphasize the present scheme's comparative merits (or potential limits).

Reviewer #1 (Remarks to the Author):

In this manuscript, the authors report a miniaturized spectrometer that can detect the spectral and polarization information with the help of trained deep neural networks. First, the authors demonstrate a 1D chirped grating, which shows distinct patterns under different illuminations. By capturing the relationship between the patterns of the microscope images and the spectra, a neural network can predict the spectra from the image patterns. Next, a 2D grating is used to resolve spectra for orthogonal polarization, and based on that optical rotary dispersion can be measured. Overall, the idea of the work is novel, the results are comprehensive, and the manuscript is written very well. I recommend publication of the manuscript in Nature Communications, after the authors address some questions and comments.

General response: We appreciate the positive feedback from this reviewer. Here we provide a detailed response to the comments.

1. I suggest the authors citing some relevant papers, including "Probabilistic Representation and Inverse Design of Metamaterials Based on a Deep Generative Model with Semi-Supervised Learning Strategy," *Adv. Mater.* (2019); "Compounding meta-atoms into metamolecules with hybrid artificial intelligence techniques," *Adv. Mater.* (2020); "Deep learning for the design of photonic structures", *Nature Photonics* 15, 77 (2021); "Pushing the limits of functionality-multiplexing capability in metasurface design based on statistical machine learning," *Adv. Mater.* (2022); "Ultraspectral Imaging Based on Metasurfaces with Freeform Shaped Meta-Atoms," *Laser & Photonics Reviews* 16, 2100663 (2022).

Response:

We appreciate this comment. The first four studies listed by the reviewer discuss methods of achieving the inverse design of metasurfaces by specifying the optical properties and behaviors desired by the end user. Each of these studies approach this challenge by using some facet of artificial intelligence or machine learning. The last article listed discusses a method of using unconventional geometries for creating meta-atoms that exhibit high spatial and spectral resolution for ultraspectral imaging. These articles are highly related to our present work. Therefore, we agree with the reviewer that including citations of these articles will significantly improve the strength of our literature survey and better reflect the novelty of our work in comparison to recent publications.

In this revision, we updated the list of references to include the articles listed by the reviewer. These articles are listed and numbered as follows:

13. Ma, W. et al. Deep learning for the design of photonic structures. *Nature Photonics* 15, 77-90 (2021).
15. Yang, J. et al. Ultraspectral Imaging Based on Metasurfaces with Freeform Shaped Meta-Atoms. *Laser & Photonics Reviews* 16, 2100663-n/a (2022).
22. Ma, W., Cheng, F., Xu, Y., Wen, Q. & Liu, Y. Probabilistic Representation and Inverse Design of Metamaterials Based on a Deep Generative Model with Semi-Supervised Learning Strategy. *Advanced Materials (Weinheim)* 31, e1901111-n/a (2019).
23. Liu, Z. et al. Compounding Meta-Atoms into Metamolecules with Hybrid Artificial Intelligence Techniques. *Advanced Materials (Weinheim)* 32, e1904790-n/a (2020).

24. Ma, W. et al. Pushing the Limits of Functionality-Multiplexing Capability in Metasurface Design Based on Statistical Machine Learning. *Advanced Materials (Weinheim)* 34, e2110022-n/a (2022).

The numbering of the references and citations were also updated to reflect these changes.

2. I suggest the authors making more detailed comparisons between the current work and the literature (including but not limited to Ref. 4-6 and "Ultraspectral Imaging Based on Metasurfaces with Freeform Shaped Meta-Atoms." *Laser & Photonics Reviews* 16, 2100663 (2022)), regarding the approach/design, device size, spectral resolution, realized functionalities, etc. A table would be helpful.

Response:

We totally agree with the reviewer that it would be helpful to include a list of comparisons between our presented work and those described in other studies to highlight the advantages and limitations of using our proposed scheme. In this revision, we included the table below highlighting details of our proposed scheme and others from several references. Please refer to line 9-10 on page 10 and **Note S17** in the supporting information.

Table R1. Comparative figures-of-merit (FOMs) of the scheme proposed in this paper and schemes from several references. *CV stands for Coefficient of Variance, which is defined by the referenced article as the ratio of the standard deviation to the mean of the photocurrent measurements. **DOP is defined by the reference as Degree of Polarization. ***Errors are listed as the average peak localization error, bandwidth error, height error, and MSE of the reconstructions, respectively. ****Data for this block was not included in the reference.

Reference	Design	Fabrication	Device Size	Resolution	Operational Range	Error	Functionality
This work	2D plasmonic chirped grating on metal film	Metal deposition, focus ion-beam milling	87.95 × 87.95 μm ²	0.2 nm (90% accuracy) 0.5 nm (95% accuracy)	470 – 770 nm	0.0026 ± 0.0013 NMSE	Spectral reconstruction, polarization reconstruction
[2]	195 colloidal quantum dot filters coupled to a camera	CQD growth, printing of CQD/PVD solution, integration onto CCD	8.5 × 6.8 mm	2 nm	390 – 690 nm	0.022 std. dev.	Spectral reconstruction
[3]	Single compositionally engineered nanowire	Annealing, e-beam lithography, plasma treatment, atomic layer deposition, metal deposition and liftoff	0.5 × 75 μm	10 nm	500 – 630 nm	2% CV*	Spectral imaging, spectral reconstruction
[4]	Photonic crystal slabs on CMOS sensors	E-beam lithography, reactive ion etching	210 × 210 μm	≈ 1 nm	550 – 750 nm	< 0.05 MSE	Hyperspectral imaging, spectral reconstruction
[8]	Photonic crystal structure coupled with photodetectors	Deep UV lithography, plasma etching	1 × 0.3 mm	-****	-90° – +90°	0.07 DOP std. deviation**	Polarization analysis
[9]	Compressive spectroscopy via thin film filter array	E-beam deposition, film deposition, photolithography	2.5 × 2.5 mm	-****	500 – 1000 nm	0.016 MSE	Spectral reconstruction
[15]	Metasurfaces composed of	E-beam lithography, ICP etching, wet	8 × 6.4 mm	0.5 nm	450 – 750 nm	0.024 nm std. dev.	Spectral reconstruction,

	freeform shaped meta-atoms	etching, PDMS transfer					ultraspectral imaging
[16]	Plasmonic spectral encoder chip comprised of 252 nanohole arrays	E-beam lithography, imprint molding, e-beam evaporation	4.8×3.6 mm	-****	480 – 750 nm	0.19 nm 0.18 nm 7.60% 7.77×10^{-5} ***	Spectral reconstruction

3. On page 5 line 9, the authors mentioned ‘A fully connected network with 4 convolutional layers’. People typically distinguish the terms convolutional neural networks (CNNs) and fully connected neural networks (FCNN) since they have different configurations and data flows.

Response: We apologize for this unclear sentence. We agree with the reviewer that convolutional neural networks (CNNs) and fully connected neural networks (FCNN) are different terms. In most CNNs, there are some fully connected layers after several convolutional layers. Our learning network (**Fig.2**) is also the case. To clarify this point, we have updated the sentence to “a convolutional neural network with 4 convolutional layers and 2 fully connected layers”. Please see line 22 on page 5.

4. In the first part of the work, the authors use a 1D chirped grating. If the length of the groove or grating is very long and fabrication is nearly perfect, the pattern should only change in the horizontal direction (where the period is changing), maintaining nearly the same value in the vertical direction. Is this the case in this work? If so, it might be advantageous to average the image along the vertical axis to convert the image from a two-dimensional (2D input) format to a one-dimensional (1D) format. The neural network can then be built using an FCNN (or 1D CNN), which will be more efficient in terms of training and prediction.

Response: We appreciate this insightful comment and agree with the reviewer that the neural network can be built using an FCNN when we solve 1D chirped gratings. Just as the reviewer said, “If the length of the groove or grating is very long and fabrication is nearly perfect, the pattern should only change in the horizontal direction”. However, the fabrication and the imaging process are not perfect. The pattern slightly changes along the vertical direction and the pattern might also be slightly slanted in the image (as seen in the 90-degree rotated image in Fig. R1(a)). As a result, averaging over the vertical direction will have a negative effect if we change the two-dimensional (2D) input to a one-dimensional (1D) input for the network.

To demonstrate that 2D CNN has superior behavior, we built an FCNN for the averaged 1D signal as shown in **Fig.R1(a)**. The FCNN includes 3 dense layers whose sizes are 256, 512, and 600, respectively. The same data as 2D CNN were used for training and testing. Among those data, 500 paired data were used for training and the rest for testing. **Figs. R1(c)** and **R1(d)** compare the spectral estimation results for both the 1D FCNN network and the proposed 2D CNN network for two sets of testing data (peak wavelengths at 460 nm + 635 nm and 470 nm + 595 nm + 770 nm, respectively).

We also show the mean and standard deviation of the NMSEs of the 2D FCNN and 2D CNN results in **Fig.R1(b)**, which are based on a total of 100 testing images. The colored bar is the mean value of each result. The error bar indicates the standard deviation. We evaluated the reconstruction accuracy using different amounts of training data (20%-80%). All these quantitative metrics indicate that 2D CNN provides more accurate spectrum reconstruction.

Based on this reason, we did not change the two-dimensional (2D) input to a one-dimensional (1D) input for the network in this revision.

Figure R1 (a) The architecture of the FCNN network. (b) Comparison of the NMSEs of the reconstructed spectra using the FCNN and the CNN. For the cases of (c) two-peak and (d) three-peak wavelength combinations, comparison of CNN-reconstructed spectrum (solid blue line), FCNN-reconstructed spectrum (solid orange line), and the gold standard spectrum measured by a conventional spectrometer (dotted red line).

5. The so-called one-to-many mapping issue often exists in the inverse design problem. For instance, very similar or almost identical spectra maybe produced by distinct structures. This issue causes challenges in the inverse design. Does the current study encounter the one-to-many mapping issue? In other words, do apparently dissimilar spectra result in a very similar or almost identical pattern (limited by the resolution of the camera)? If so, what approach did the authors take to resolve or mitigate the issue?

Response: We appreciate this great comment for us to emphasize the advantage of the proposed spectroscopic chip. The rainbow trapping structure is able to produce a unique one-to-one pattern because of the physical nature of the graded grating. According to our earlier fundamental investigation of the graded grating (i.e. ref. [29-31]), the light trapping position corresponds to a single wavelength only. For

instance, each narrow band incident wavelength shown in Fig. 1 resulted in a unique pattern. In our investigated spectral range, there is no one-to-many mapping issue. This is one of the advantages of the proposed rainbow trapping structure. In this revision, we emphasized this advantage in line 22-23 on page 3.

6. The present work is in reflection configuration rather than transmission configuration. Is there a special reason, and what are the possible challenges in transmission-type devices using chirped gratings?

Response: We appreciate this comment. It is possible to design the smart ‘rainbow’ spectrometer system using a transmission configuration. However, using a reflection configuration introduces a simpler design process. When designing a grating, the groove depth is one of the main contributing factors for determining its coupling efficiency. However, transmission configurations require thin metal films (few 10s of nm) to minimize light absorption and reflection. Thus, fabricating the ‘rainbow’ chip in the transmission configuration presents a challenge in finding a film thickness that optimizes the balance between the transmittance and the coupling efficiency of the surface rainbow. Conversely, using a reflection configuration requires thicker films (> 100 nm) to reduce transmission, simplifying the design process to an optimization of the groove depth. But we do agree with the reviewer that transmission mode can result in a more compact optical system, which is still under investigation.

Reviewer #2 (Remarks to the Author):

Efforts to miniturize spectroscopic systems are currently a very timely area - but such miniturization usually leads to inaccurately reconstructed spectra. Attaining accurate reconstruction by simply exploiting images of spectral resonances is a key current and ongoing objective of the broader field of optical spectroscopy. What this paper reports is a novel artificial-intelligence based method for obtaining accurate such spectra - for miniturized devices. Although the training time for the AI method may be long, the time for spectrum-reconstruction is efficient once the overall model is calibrated.

The 'rainbow trapping' structure that the authors use for obtaining the images has been optimized and pioneered by the authors over the years (e.g., Refs. 24-26); it allows for localizing different incident light-wavelengths to different spatial points along the structure (a chirped plasmonic grating), thereby being very broadband and obviating the need for (many slightly different) resonant filters.

Overall, this is a well-written, high-quality work, combining nicely several timely areas (slow light / 'rainbow trapping', optical spectroscopy and polarimetry, and AI-based systems) to attain a useful functionality (described above).

As such, it makes an interesting / novel contribution to the broader field, meriting publication in NCOMMs after, from my perspective, the authors also address the following points:

General Response: We appreciate the positive feedback from this reviewer. Here we provide a detailed response to the comments.

1. Is the used AI method guaranteed to converge to a global (rather than 'only' local) error-minimum point? That would be interesting to know, in order to assess the generality and general-reliability of the method (here, it does appear to be ok).

Response: We appreciate this comment. Like most deep learning methods, our proposed AI-based method cannot guarantee to converge to a global error-minimum point. Gradient descent converges to a local minimum. If there are multiple local minimums, its convergence depends on where the iteration starts. Only if the loss function is convex and the step size is chosen with a condition that the function strictly decreases, then the gradient descent algorithm converges to the global minimum. Therefore, convergence to a global minimum cannot be guaranteed in general. However, for our proposed method, the local minimum is reasonably good in Fig 2 of the main text. Specifically, as shown in **Fig. R2**, we plot the accuracy rate and loss of training data during 1D spectrum reconstruction training (data parameters are shown in **Table S1**). The accuracy is defined if the peak of reconstruction and measure are in the same position. The loss was calculated using mean-squared error (MSE):

$$Loss = \sum_{i=1}^n \|\hat{x}_i - x_i\|_2^2 \quad i=1, 2, \dots, n \quad (\text{R1})$$

\hat{x}_i is the reconstruction of the training patterns using the trained network, x_i is the training spectra obtained from the experiment, and n is the number of training data. An epoch means training the neural network with all the training data for one cycle. A total of 400 epochs were used in our network. We can see the accuracy increases and the loss decreases during training. This makes sure our gradient descent converges to a local

minimum. However, our proposed DL-based method cannot guarantee to converge to a global error-minimum point like all other DL methods.

Fig. R2. The accuracy and loss of training data during training.

2. What is the maximum spectral resolution that can be attained with this scheme (on the reported structure)? Also, what is here the (well-known) figure-of-merit for the reported spectro-polarimeter?

Response: We appreciate this insightful comment. Spectral resolution is one of the most important parameters to evaluate the performance for conventional spectrometers. Here we explored two spectral resolutions: i.e., (1) the resolution in wavelength shift, and (2) the resolution to resolve two adjacent peaks.

(1) The resolution in wavelength shift: Here we employed a broadband halogen lamp through a liquid crystal filter to reveal its resolution in wavelength shift. We first captured 10,000 images of the rainbow chip under the illumination of narrowband incidence from 600 nm to 650 nm with the step size of 0.1 nm tuned by the liquid crystal filter. Their actual spectra were characterized using the fiber-based spectrometer. 8,000 (and 9,000) images have been selected randomly as training data. After training, we tested the remaining 2,000 (and 1,000) images which were not included in the training data. As shown in **Figs. 3a** and **3b**, single peaks can be reconstructed and well resolved with the peak shift of 0.5 nm (Fig. 3a) and 0.2 nm (Fig. 3b). The accuracy of the reconstructed peak position is 87% - 95% for the peak shift of 0.5 nm, and 81% - 90% for the peak shift of 0.2 nm (see **Table S3**). More technical details to resolve wavelength shifts with different step sizes are listed in **Note S6**.

(2) The resolution to resolve two adjacent peaks: To further reveal the spectral analysis capability, we then introduced two narrow peaks controlled by a programmable acoustic optical filter to illuminate the grating simultaneously. Seven representative spectra of the incident narrowband light are plotted in **Fig. 3c**: One peak was fixed at the wavelength of 596.8 nm. The other narrow peak was tuned from 596.8 nm to 646.8 nm with the step size of 0.1 nm. As shown by spheres in **Fig. 3d**, these two adjacent incident peaks produced a combined spectrum, showing that the two peaks gradually separate apart with each other and therefore can be resolved by the conventional spectrometer. In this experiment, we collected 901 images as the training set and 100 images for testing (see details in **Note S7**). The reconstructed spectra are plotted by solid curves in **Fig. 3d**, agreeing perfectly with the measured spectra. One can see that the two peak identification is similar to determine the optical resolution in imaging applications using the Rayleigh criterion. According to our reconstructed and measured spectra, the two-peak feature was clearly resolved

when the wavelength difference is beyond 2 nm (see detailed analysis in *Note S8*). These preliminary data indicated the potential using the smart rainbow chip system to perform high resolution spectral analysis with the equivalent performance compared with conventional spectrometers.

In this revision, we added a new Fig. 3 to explain the resolution performance of the proposed smart spectrometer on a chip. Please refer to the highlighted paragraphs on page 6-7.

Figure 3| The resolution of the smart system. (a-b) DL-reconstructed spectrum (solid lines) with a step size of 0.5 nm (Fig. 2a) and 0.2 nm (Fig. 2b), respectively, and the measured spectra using a conventional spectrometer (spheres). The peak positions are indicated by vertical dashed lines [i.e., 605.0 nm (blue line), 605.5 nm (green line) and 606.0 nm (red line) in Fig. 3a, and 605.2 nm (blue line), 605.4 nm (green line) and 605.6 nm (red line) in Fig. 3b]. (c) The spectra of the two incident-light measured independently. (d) DL-reconstructed spectrum (solid lines) and the measured spectra of the two combined peaks using a conventional spectrometer (spheres).

As for the smart spectro-polarimeter, typical figure-of-merits (FOM) include angular resolution and the polarization range. In our reported data, these two FOMs are determined by the training data set. In Table S9 in the supporting information, we analyzed the deviations of the reconstructed angles at different wavelengths, ranging from 0.07 – 0.45. This limitation is a technical issue which can be improved using finely tuned electronic-driven polarizers to produce training datasets for future studies. In this revision, we discussed this relatively large deviation in line 42-50 on page 9 and line 1-10 on page 10.

3. How much susceptible (or not) to noise is the scheme, and why?

Response: This is a very good question regarding the resilience to noise of our proposed system. As with most CNN-based methods, our proposed method is rather robust to noise. The CNN intrinsically has denoising capability and has been used widely for denoising images. As can be seen in the manuscript, the noise level is pretty high in the images as we used low-cost optics system and components. Our method is still able to reconstruct the spectrum accurately. On the other hand, increased noise will negatively affect the accuracy of the spectrum reconstruction for the proposed method. For instance, the noise is higher in the 2D grating for spectro-polarimeter experiment shown in Fig. 5 in the revision. As a result, the error is obviously larger than the 2-peak and 3-peak spectra reconstruction shown in Fig. 2. It is still under investigation to improve the quality of the 2D grating to enhance the signal-to-noise ratio.

4. It should be interesting for the journal's broad readership to compare this scheme with other standard schemes / papers (e.g., cf. Refs. [1-4]) for spectral and polarimetric analyses (e.g., in terms of fabrication requirements, scalability, etc) in order to even more clearly outline and emphasize the present scheme's comparative merits (or potential limits).

Response: Both reviewers asked this insightful question. As such, we present the details of our proposed scheme in comparison with other papers in the table below. In this revision, we included the table below highlighting details of our proposed scheme and others from several references. Please refer to line 9-10 on page 10 and Note S17 in the supporting information.

Table R1. Comparative figures-of-merit (FOMs) of the scheme proposed in this paper and schemes from several references. *CV stands for Coefficient of Variance, which is defined by the referenced article as the ratio of the standard deviation to the mean of the photocurrent measurements. **DOP is defined by the reference as Degree of Polarization. ***Errors are listed as the average peak localization error, bandwidth error, height error, and MSE of the reconstructions, respectively. ****Data for this block was not included in the reference.

Reference	Design	Fabrication	Device Size	Resolution	Operational Range	Error	Functionality
This work	2D plasmonic chirped grating on metal film	Metal deposition, focus ion-beam milling	$87.95 \times 87.95 \mu\text{m}^2$	0.2 nm (90% accuracy) 0.5 nm (95% accuracy)	470 – 770 nm	0.0026 ± 0.0013 NMSE	Spectral reconstruction, polarization reconstruction
[2]	195 colloidal quantum dot filters coupled to a camera	CQD growth, printing of CQD/PVD solution, integration onto CCD	$8.5 \times 6.8 \text{ mm}$	2 nm	390 – 690 nm	0.022 std. dev.	Spectral reconstruction
[3]	Single compositionally engineered nanowire	Annealing, e-beam lithography, plasma treatment, atomic layer deposition, metal deposition and liftoff	$0.5 \times 75 \mu\text{m}$	10 nm	500 – 630 nm	2% CV*	Spectral imaging, spectral reconstruction
[4]	Photonic crystal slabs on CMOS sensors	E-beam lithography, reactive ion etching	$210 \times 210 \mu\text{m}$	$\approx 1 \text{ nm}$	550 – 750 nm	< 0.05 MSE	Hyperspectral imaging, spectral reconstruction
[8]	Photonic crystal structure coupled with photodetectors	Deep UV lithography, plasma etching	$1 \times 0.3 \text{ mm}$	-****	$-90^\circ - +90^\circ$	0.07 DOP std. deviation**	Polarization analysis

[9]	Compressive spectroscopy via thin film filter array	E-beam deposition, film deposition, photolithography	2.5 × 2.5 mm	-****	500 – 1000 nm	0.016 MSE	Spectral reconstruction
[15]	Metasurfaces composed of freeform shaped meta-atoms	E-beam lithography, ICP etching, wet etching, PDMS transfer	8 × 6.4 mm	0.5 nm	450 – 750 nm	0.024 nm std. dev.	Spectral reconstruction, ultraspectral imaging
[16]	Plasmonic spectral encoder chip comprised of 252 nanohole arrays	E-beam lithography, imprint molding, e-beam evaporation	4.8 × 3.6 mm	-****	480 – 750 nm	0.19 nm 0.18 nm 7.60% 7.77×10^{-5} ***	Spectral reconstruction

REVIEWERS' COMMENTS

Reviewer #1 (Remarks to the Author):

The authors have successfully addressed my comments, and revised the manuscript accordingly. I recommend the acceptance of the manuscript as is.

Reviewer #2 (Remarks to the Author):

I have read again the updated version of the manuscript, and the authors' replies to both reviewers.

The initial / submitted version was already in very good shape, and the present / revised work (+ SI) has substantially improved, with the authors clearly explaining (for a broad readership), among others, the merits of the reported device in terms of footprint, spectral resolution, noise performance and functionality compared with previous 'similar' devices.

As such, from my perspective, this interesting and timely work, now further improved, merits publication in the journal as is.